# Quality of Life after Mandibular Reconstruction Using Free Fibula Flap and Customized Plates: A Case Series and Comparison with the Literature

**DOI:** 10.3390/cancers15092582

**Published:** 2023-04-30

**Authors:** Jorge Pamias-Romero, Manel Saez-Barba, Alba de-Pablo-García-Cuenca, Pablo Vaquero-Martínez, Joan Masnou-Pratdesaba, Coro Bescós-Atín

**Affiliations:** 1Service of Oral and Maxillofacial Surgery, Hospital Universitari Vall d’Hebron, Vall d’Hebron Barcelona Hospital Campus, Passeig Vall d’Hebron 119-129, E-08035 Barcelona, Spain; jorge.pamias@vallhebron.cat (J.P.-R.); manuel.saez@vallhebron.cat (M.S.-B.); alba.depablo@vallhebron.cat (A.d.-P.-G.-C.); pablo.vaquero@vallhebron.cat (P.V.-M.); 2CIBBM-Nanomedicine, Noves Tecnologies i Microcirurgia Craniofacial, Vall d’Hebron Institut de Reserca (VHIR), Hospital Universitari Vall d’Hebron, Vall d’Hebron Barcelona Hospital Campus, E-08035 Barcelona, Spain; 3Radiology Department, Hospital Universitari Vall d’Hebron, Vall d’Hebron Barcelona Hospital Campus, Passeig Vall d’Hebron 119-129, E-08035 Barcelona, Spain; joan.masnou@vallhebron.cat; 4Unitat Docent Vall d’Hebron, Facultat de Medicina UAB, Universitat Autònoma de Barcelona, E-08035 Barcelona, Spain

**Keywords:** quality of life, mandibular reconstruction, free fibula flap, patient-specific implant, plates, University of Washington Quality of Life Questionnaire

## Abstract

**Simple Summary:**

The health-related quality of life was evaluated in 23 patients undergoing mandibular reconstruction with free fibula flap and titanium customized plates. A computer-aided design and computer-aided manufacturing technology were used. The University of Washington Quality of Life questionnaire for head and neck cancer patients is a widely used and validated tool, which was self-completed by the patients after 12 months of surgery. In the 12 single question domains, the highest scores were obtained in the domains of taste, shoulder function, anxiety, and pain. The lowest scores corresponded to chewing, appearance, saliva, and mood. The global quality of life was rated as good, very good, or outstanding by 81% of patients. The present results compared favorably with previous studies of mandibular reconstruction using the same questionnaire published in literature.

**Abstract:**

A single-center retrospective study was conducted to assess health-related quality of life (HRQoL) in 23 consecutive patients undergoing mandibular reconstruction using the computer-aided design (CAD) and computer-aided manufacturing (CAM) technology, free fibula flap, and titanium patient-specific implants (PSIs). HRQoL was evaluated after at least 12 months of surgery using the University of Washington Quality of Life (UW-QOL) questionnaire for head and neck cancer patients. In the 12 single question domains, the highest mean scores were found for “taste” (92.9), “shoulder” (90.9), “anxiety” (87.5), and “pain” (86.4), whereas the lowest scores were observed for “chewing” (57.1), “appearance” (67.9), and “saliva” (78.1). In the three global questions of the UW-QOL questionnaire, 80% of patients considered that their HRQoL was as good as or even better than it was compared to their HRQoL before cancer, and only 20% reported that their HRQoL had worsened after the presence of the disease. Overall QoL during the past 7 days was rated as good, very good or outstanding by 81% of patients, respectively. No patient reported poor or very poor QoL. In the present study, restoring mandibular continuity with free fibula flap and patient-specific titanium implants designed with the CAD-CAM technology improved HRQoL.

## 1. Introduction

Refinements in surgical techniques have led to significant improvement in oncological, functional, and aesthetic outcomes in oral cancer. Currently, one of the main goals of mandibular defect reconstruction is to provide patients with the best possible health-related quality of life (HRQoL) [1]. Assessment of results of treatment is a key aspect for the accurate selection of patients and the choice of the most appropriate reconstruction technique [2,3].

The microvascular or free fibula flap, originally described by Hidalgo et al. [4] in 1989, is considered the “gold standard” flap for the reconstruction of mandibular defects. More recently, the use of computer-aided design (CAD) and computer-aided manufacturing (CAM) (CAD-CAM) technology [5] promoted a paradigm shift in the diagnostic and therapeutic approach of defects in the maxillofacial region. Further introduction of 3D-printed titanium using direct metal laser sintering (DMLS) as an additive manufacturing technique allowed the development of custom-made plates or patient-specific implants (PSIs), improving accuracy and efficiency in mandibular reconstruction procedures. PSIs could provide the missing link in the digital flow process for mandibular reconstruction, and in doing so they would avoid potential shortcomings that are inherent to pre-modelled reconstruction plates and improve final precision [6,7]. Thus, computer-generated PSIs would be the next logical step in the digital planning and design flow rather than an independent device, as they represent the metallic cast that accurately reflects the surface of the reconstructed bone compounds and keeps geometry stable [8].

The evidence of PSI printed titanium implants for reconstruction of mandibular continuity defects is scarce. In a systematic review of the literature of 31 clinical studies with 139 patients, benefits identified included finite element analysis of the digital design, dimensional accuracy, shorter duration of surgery, augmenting dental/masticatory function, and capacity for dental implant rehabilitation, although the evidence predominantly was low level and at moderate-to-high risk of bias [9]. The published articles provided valuable evidence of the use of 3D-printed titanium PSIs with reported benefits seemingly outweighing their limitations and of the important role to be played by such implants in mandibular reconstruction for improving patient outcomes. However, in none of the studies included in the review was HRQoL evaluated. Improvements in different domains of HRQoL and patient satisfaction after free fibula flap reconstruction of segmental mandibulectomy have been rarely reported [10,11,12,13,14,15], but as far as we are aware, no studies have specifically assessed HRQoL outcomes in the setting of mandibular reconstruction using free fibula flaps combined with 3D-customized titanium plates.

Therefore, the aim of this study was to assess the impact of using free fibula flaps associated with CAD-CAM technology and PSI titanium plates on HRQoL in patients with mandibular pathology undergoing reconstruction for continuity defects.

## 2. Materials and Methods

### 2.1. Study Design and Participants

This was a retrospective study of all consecutive patients undergoing mandibular reconstruction with free fibula flaps using CAD-CAM technology and titanium PSI for the repair of mandibular defects of malignant or benign etiology operated on at the Service of Oral and Maxillofacial Surgery of Hospital Universitari Vall d’Hebron in Barcelona (Spain) between October 2015 and July 2019. Inclusion criteria were as follows: adult patients scheduled for primary or secondary mandibular reconstruction due to benign or malignant pathology, whether diseases had been treated previously or not; use of CAD-CAM technology including virtual planning, mandibular resection, fibula cutting guides for modelling, and PSI; use of free fibula flap for the reconstruction of the mandibular bone defect; and follow-up for at least 1 year after surgery. Patients were excluded if one or several components of the CAD-CAM technology were lacking (such as virtual planning, mandibular resection guides, fibula cutting guides for modelling), PSI was not used, or the free fibula flap failed.

The study protocol was approved by the Clinical Research Ethics Committee of Hospital Universitari Vall d’Hebron (codes PR(AG)93/2016, approval date 1 March 2016) (Barcelona, Spain). Written informed consent was obtained from all participants.

### 2.2. Protocol for Mandibular Reconstruction with Free Fibula Flap

Briefly, the presurgical stage included the following steps: (a) virtual planning (image processing, segmentation, resection, cutting, and reconstruction planning); (b) CAD (mandibular resection guides, fibula cutting guides, custom-made reconstruction plates, and custom-made prostheses); and (c) manufacturing stage (polyamide models from StereoLitography (STL) file format for resection and cutting guides for the mandible and fibula, STL model for the mandible, 3D printing and manufacturing titanium plates [PSI], and custom-made polyetheretherketone [PEEK] prosthesis) (Figure 1). Custom-made plates were manufactured using direct metal laser sintering using an EOSINT M270 system (EOS GmbH, Electro Optical Systems Company, Munich, Germany).

The surgical procedure (Figure 2) included the following steps: (1) mandibular resection using resection guides; (2) modelling of the fibula flap using cutting guides and placement of immediate implants (if required); (3) plate binding in the donor zone before sectioning the vascular pedicle; (4) positioning and binding of the flap in the mandibular defect; (5) microsurgical anastomosis; (6) positioning and binding of the PEEK prosthesis with miniplates and screws (if required); and (7) final repositioning of soft tissues and wound closure. All plates were customized for each patient. In all cases, PSI modelling was performed in the limb while the flap remained vascularized.

Anatomical models, surgical guides, and custom-made plates were designed using the specific design software “D-matic Medical ^®^ 10.0 by Materialise”. Biomodels were manufactured directly using a rapid prototyping machine that used tridimensional solid support technology (Stratasys, Eden Prairie, MN, USA). Plates were manufactured using direct sintering with metal laser using an EOSINT M270 system (Electro-Optical Systems, GmbH, Munich, Germany).

### 2.3. Evaluation and Follow-up

Patients were visited postoperatively by the same investigator (J.P.-R.) during their stay in the hospital, after 1 week of hospital discharge, and at 1, 3, 6, and 12 months thereafter. Postoperative complications were evaluated using the Clavien–Dindo classification [16]. Complications related to PSI (presence or absence of intraoral or extraoral exposure) and the PEEK prosthesis (stability) were evaluated clinically. Prosthesis failure was determined when the prosthesis was extra-orally exposed and had to be removed. Other variables were evaluated by orthopantomography and computed tomography (CT) scan performed at least 6 months after mandibular reconstruction, including merging of fibula fragments (between different fibula fragments and between the fibula fragments and the remaining mandible), stability of screws, plate adjustment (defined as the presence of close contact between the PSI, the fibula, and the mandible), and presence or absence of PSI fracture.

Esthetical evaluation included photographs of the patients before and after surgery. Additionally, pre- and post-surgical panoramic radiographs and 3D-cone-beam computed tomography (CBCT) scans were acquired, and image superposition was used to assess the correlation between virtual planning and the results obtained.

### 2.4. Health-Related Quality of Life

At least 12 months after surgery, patients were contacted by phone and were appointed for a face-to-face visit to assess HRQoL. After signing the informed consent, they completed the University of Washington Quality of Life Questionnaire (UW-QOL v4) for head and neck cancer patients [17,18]. A Spanish validation version of the UW-QOL instrument was used [19]. The UW-QOL is a self-administered questionnaire specifically for head and neck cancer patients that measures health and quality of life (QoL) over the past 7 days. The questionnaire includes 12 single question domains (pain, appearance, activity, recreation, swallowing, chewing, speech, shoulder function, taste, saliva, mood, and anxiety) and 3 global questions, one about how patients feel relative to before they developed their cancer, one about their HRQoL, and one about their overall QoL. A free-text box is also included, so that the patient may write down any other comment he or she wishes to make on QoL that had not come forth in the previous questions. Domains are scaled from 0 (worst possible response) to 100 (best possible response). Domain scores include the mean (SD), the percentage of patients selecting the best possible response (100), and the percentage of patients choosing each domain. The domains can also be ranked by order.

### 2.5. Study Outcomes

The primary outcome was HRQoL assessed by means of the UW-QOL questionnaire at least 12 months after mandibular reconstruction using free fibula flap and titanium PSI based on the CAD-CAM technology. Secondary outcomes were complications related to the PSI and the PEEK prosthesis.

### 2.6. Statistical Analysis

Categorical variables are expressed as frequencies and percentages, and continuous variables are expressed as mean and standard deviation (SD) or median and interquartile range (IQR) (25th–75th percentile) or range (maximum–minimum). The chi-square test or the Fisher’s exact test were used for the comparison of categorical variables, and the Student’s *t*-test or the Mann–Whitney *U* test were used for the comparison of quantitative variables according to conditions of application. Statistical significance was set at *p* < 0.05.

## 3. Results

### 3.1. Clinical and Surgical Characteristics

The study population consisted of 23 patients (56.5% men) with a mean age of 52.8 (14.2) years. Fifteen patients (65.2%) had malignant tumors and locally advanced disease. Four patients had received neoadjuvant radiotherapy or combined radiochemotherapy.

Central defects according to the classification of Boyd et al. [20] were the most common (56.5%). PSIs were inserted in the occlusal zone in 15 patients and in the basal zone in the remaining 8. The skin flap was used as an internal intraoral layer in 20 patients, as an external skin layer in 2, and both as internal and external layers in 1. One patient required bilateral nasolabial flaps because of a defect that involved a large amount of soft tissue. Arterial anastomosis was most frequently performed with the facial artery and venous anastomosis with the thyrolinguofacial trunk. Osseointegrated dental implants were placed immediately in 2 patients and in a second step in 3.

The mean (SD) ischemia time was 122 (4) minutes, and the mean duration of surgery was 10.2 (1.4) hours. Immediate postoperative complications were recorded in 11 patients, which were classified as grade I in 7 and grade IIIb in 4 (2 cases of cervical bleeding and 2 of compartment syndromes in the donor limb). These 4 patients were reoperated under general or local anesthesia. In all cases, complications were solved. The mean length of hospital stay was 23 days (range 10–55 days), without significant differences between patients without and with complications (17 [4.4] vs. 26.3 [12.9] days, *p* = 0.062).

The microvascular fibula flap survived in 100% of the patients. Postoperatively, 12 patients received chemotherapy and/or radiotherapy adjuvant treatment. Table 1 shows the main clinical characteristics of patients and surgery-related data.

Image superposition studies showed a high correlation (greater than 92% in most patients) between preoperative virtual surgical plan and the results obtained.

The mean length of follow-up was 26 months (range 12–50 months). Twenty-two patients (95.6%) were alive at 12 months after surgery. One patient developed a recurrence of their oral cancer and the other patient died due to cancer progression.

### 3.2. Health-Related Quality of Life

Twenty-one patients (91.3%) completed the UW-QOL questionnaire, after a median of 27 months (IQR 19–41 months) after primary surgery. Two patients did not complete the questionnaire; one patient had an advanced stage of the oral cancer due to recurrence, and the other patient had died.

Table 2 shows the results obtained in the 12 single question domains of the UW-QOL questionnaire. The highest mean scores were found for “taste” (92.9 [13.1]), “shoulder” (90.9 [18.4]), “anxiety” (87.5 [24.5]), and “pain” (86.4 [12.8]). In contrast, the lowest mean scores were observed in the domains of “chewing” (57.1 [39.6]), “appearance” (67.9 [19.6]), and “saliva” (78.1 [27.3]).

The highest percentages of patients selecting the best possible response (100) were 76% for “shoulder” and “taste”, 70% for “anxiety”, 67% for “swallowing”, and 52% for “recreation” and “saliva”. The lowest percentages corresponded to 10% for “appearance”, 38% for “chewing”, and 43% for “activity”.

In relation to importance of domain, “chewing”, “appearance”, and “speech” were selected by 62%, 48%, and 43% of patients, respectively. “Recreation” and “shoulder” were chosen by only 5% of patients, respectively. The rank order of domains was consistent with the importance already assigned to the different domains.

In the three global questions of the UW-QOL questionnaire (Table 3), 80% of patients considered that their HRQoL was as good as or even better than it was compared with their HRQoL before cancer, and only 20% reported that their HRQoL had worsened after the presence of the disease. Additionally, HRQoL and overall QoL during the past 7 days were rated as good, very good, or outstanding by 81% of patients, respectively. No patient reported poor or very poor QoL.

Thirteen patients (61.9%) provided an answer in the free-text box of the questionnaire. Four patients explicitly stated their satisfaction with the outcomes of surgery, but 9 patients would like to undergo dental rehabilitation for improving chewing and aesthetic functions. Other complaints were the possibility of a secondary reconstruction to improve appearance (3 cases), reduction in the extension of mouth opening (1 case), decreased saliva output and taste alterations (1 case), paresthesia (1 case), and delayed wound healing and/or paresthesia in the graft area of the lower limb.

### 3.3. PSI-Related Complications

At 6 months after surgery, 22 out of 23 patients (95.6%) underwent clinical and radiological assessment. One patient moved to another city and was lost to follow-up. The fibula fragments were properly consolidated in all 22 patients. In 19 patients (86.4%), PSI-related complications did not occur, whereas complications were recorded in the remaining 3 patients (13.6%). Extraoral and intraoral exposure of the PSI was clinically documented in 2 patients, and in both cases, the plate was removed, but the segments of the microvascularized fibula flap were found to be well consolidated. In the remaining patient, there was a lack of consolidation between the fibula and the remaining mandible, with screw instability and plate mobility. In this patient, removal of both the plate and the remaining segment of the mandibular ramus were performed.

### 3.4. PEEK Prosthesis-Related Complications

A PEEK prosthesis for the reconstruction of the mandibular inferior border was performed in 14 patients (60.9%) (immediate reconstruction in 13 cases and at a later stage in 1). In 6 patients (42.8%), the prosthesis became exposed and had to be removed. Five of these 6 patients had received radiotherapy (RT) in the neoadjuvant or adjuvant setting. Removal of the PEEK prosthesis was significantly more common in patients treated with RT than in those who had not received RT (83.3% vs. 16.7%, *p* = 0.031).

## 4. Discussion

This study shows that in patients undergoing extensive mandibular resection leading to wide mandibular continuity defects, the use of a surgical procedure based on CAD-CAM technology with free fibula flap and titanium PSI was associated with high scores in the UW-QOL questionnaire at least 12 months after surgery. In the 12 single question domains, mean scores were higher than 80 (with 100 being the highest possible response) in 9 domains (75%), with only 3 domains scoring below 80%. In the three global questions of the UW-QOL instrument, HRQoL before diagnosis of malignancy and overall QoL in the previous 7 days, high scores were achieved, as 80% and 81% of patients selected the options of much better and good, very good, or outstanding, respectively.

Assessment of QoL is a clinically relevant outcome in monitoring the treatment success and the sequelae of illness in patients with oral cancer. Subjective measures of health status can be evaluated by generic or disease-specific instruments, but due to the complex anatomy of the oral cavity, it is desirable to use specific HRQoL measures. These measures are more sensitive in assessing the impact of oral conditions on daily life activities. The relatively large number of questionnaires that are specific for diseases of the oral cavity (e.g., 14-item Oral Health Impact Profile [OHIP-14], Oral Impacts on Daily Performances [OIDP], Oral Health-Related Quality of Life [OHRQoL], European Organization for Research and Treatment of Cancer Head and Neck cancer questionnaire [EORTC-H&N35]) [21], underscores the fact that there is no gold standard tool. The UW-QOL instrument is one of the most used and validated questionnaires for patients with head and neck cancer and has shown good psychometric properties that have been specifically developed for this pathology [17,18]. Furthermore, the incorporation of importance-rating domains makes UW-QOL unique among head and neck cancer instruments [22,23]. The Spanish version of this questionnaire was validated by Nazar et al. [19] in 2010. In fact, the following characteristics of the UW-QOL questionnaire stand out: (1) it provides a specific “appearance” item related to disfigurement; (2) it allows for the evaluation of appearance problems through “recreation”, “anxiety”, and “mood” domains; and (3) it is quick and simple for patients to complete (it may take 5 minutes) and is easy to process.

Despite the advantages of the UW-QOL questionnaire, few studies have used this instrument for assessing HRQoL after mandibular reconstruction using free fibula flaps. In 2019, Petrovic et al. [24] conducted a systematic review of the literature and found only 6 studies in which QoL outcomes following mandible reconstruction using free fibula flap had been evaluated using the UW-QOL questionnaire. All these studies were retrospective case series. Apart from these 6 publications, we did not find any subsequent publication of the use of this questionnaire after free fibula flap reconstruction of the mandible. Therefore, the present results are compared with data reported in these 6 studies [14,15,25,26,27,28]. As shown in Table 4, mean scores obtained in our study were higher than those reported by others, except for “appearance”. Overall, “chewing” was the domain with the lowest mean values in all studies followed by “appearance”, “anxiety”, “speech”, and “swallowing”.

In relation to the domains in which the best score (of 100) was obtained, data were reported in four studies, with “pain”, “shoulder function”, “activity”, and “recreation” as those with the most favorable evaluation (Table 5).

A remarkable finding was that the “chewing” domain had the lowest score both in our study and in the 6 studies analyzed. Additionally, this domain showed a rate of importance of 62% in the present study as compared with 76.8% in the remaining studies. On the other hand, when considering the rank order assigned to the different domains, “chewing” ranked first in all studies but one (Table 6).

Chewing has been shown to score worse after segmental mandibulectomy and reconstruction using composite free tissue transfer [29]. In these patients, rehabilitation with implant-supported prosthesis appears to improve QoL outcomes [30,31,32]. In a pilot study of 10 patients of early loaded implant-supported fixed dental prosthesis following mandibular reconstruction, patient satisfaction improved significantly after dental rehabilitation as compared to mandibular reconstruction alone [33]. Dental implants were placed in only 5 patients in our series, but 9 of the 13 patients (69.2%) reported the desire to undergo dental rehabilitation for improving chewing and aesthetic functions in the free-text box. Prosthetic rehabilitation, however, should be indicated on a case-by-case basis [31]. This decision should be based on several considerations including the medical history, prognosis, comorbidities and, particularly, the patient’s desires and expectations. In addition, special attention should be paid to the surgical planning of implants, soft tissue management, and prosthodontics in order to avoid complications and achieve stable long-term results. We also believe that tests of swallowing function could help identify patients with a preserved swallowing function, which are in fact those who would benefit most from this kind of rehabilitation.

“Appearance” in the preceding 7 days was another domain selected as one of the most important by 48% of our patients, which is consistent with percentages between 49% and 67% reported in other studies [14,15,25,28]. Although “appearance” was considered an important factor, 71.4% of our patients stated in the questionnaire that their appearance had suffered slight or no changes, 19% a moderate change, and only 9.5% (2 patients) reported feeling disfigured. However, appearance did not seem to be a reason for social isolation, as “recreation” was rated as only 5% in the importance of domain and in the 7th position of the rank order. As for the overall QoL during the past 7 days, 81% reported that it was good, very good, or outstanding, and only 4 patients (20%) considered that QoL was fair. Poor or very poor ratings were not observed.

In relation to secondary outcomes, only 3 patients presented PSI-related complications, with a rate of 13.6%, which is consistent with 12.2% reported in the systematic review of Goodson et al. [9]. Plate removal was required by only 2 patients because of exposure, but no deficiencies in the consolidation process between the fibula fragment and the mandible were found.

A PEEK prosthesis was used in 14 patients for the correction of mandibular asymmetry after free fibula flap reconstruction [34]. In 6 patients (42.8%), the prosthesis was exposed and had to be removed. It should be noted that 5 of these 6 patients had received RT for the treatment of their oncological disease. Patients in whom the PEEK prosthesis was not exposed to RT did not present complications, with satisfactory aesthetic results and stability of the mandibular contour.

Esthetical evaluation was performed using pre- and post-surgical photographs, panoramic radiographs, and 3D-CBCT scans showing a high correlation between virtual surgical plan and the results obtained. Other techniques, such as cephalometric analysis and photogrammetry, were not used as the study was focused on the assessment of QoL as a primary subjective domain.

Limitations of the study include the single-center characteristics, retrospective design, and a small study population. Additionally, patients with malignant and benign conditions were included, which may have different risk factors related to QoL, particularly the use of radiation therapy and chemotherapy. However, the aim of the study was to assess the impact of the reconstructive process of the mandible (CAD-CAM, free fibula flap, and customized titanium plates) on QoL rather than the pathology itself, and in this respect, the population was homogeneous. Patients included in other series reported in the literature with which a comparison was made (Table 6) also included patients with ameloblastoma, osteoradionecrosis, and oral squamous cell carcinoma. Preoperative data of HRQoL using the same UW-QOL questionnaire was not obtained, so a within-group comparison of QoL before and after surgery was not feasible. Although only 4 patients received neoadjuvant or adjuvant RT and/or chemotherapy, the impact of this oncological treatment (e.g., impairment of salivary gland, trismus, mucositis, mouth opening limitation, etc.) was not evaluated. Other risk factors, such as oral health status, smoking, or age were not evaluated either. However, the use of a validated HRQoL instrument, such as the UW-QOL questionnaire, after a period of at least 12 months after surgery is a strength of this study. Moreover, a detailed comparison of the present findings with other studies published in the literature in which the UW-QOL questionnaire was completed by patients undergoing similar mandibular reconstruction procedures with free fibula flap is an interesting and distinctive aspect of the study.

## 5. Conclusions

Restoring mandibular continuity with free fibula flap and patient-specific titanium implants designed with the CAD-CAM technology improved HRQoL. High scores in most specific domains of the UW-QOL questionnaire were obtained at 12 months after surgery, except for “chewing” which had the lowest score. The global QoL was considered good, very good, or outstanding by 81% of the patients. Further studies with a larger study population are necessary to confirm the present findings.

## Figures and Tables

**Figure 1 cancers-15-02582-f001:**
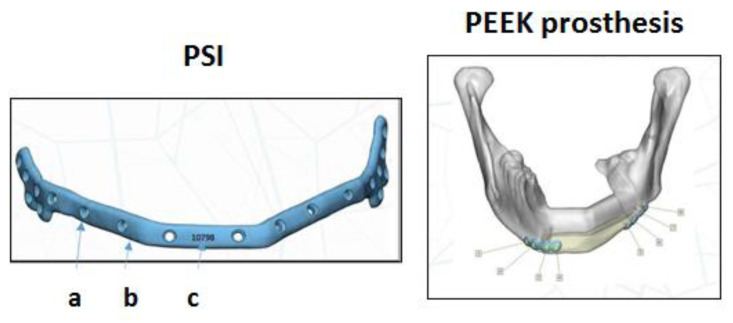
Patient-specific implant (PSI): (**a**) screw hole with thread; it contains information on the screw angle; (**b**) an enveloping design to help place the plate in the optimal position; and (**c**) patient’s information code (left). PEEK prosthesis. Positioning of the PSI in the remaining healthy bone (right).

**Figure 2 cancers-15-02582-f002:**
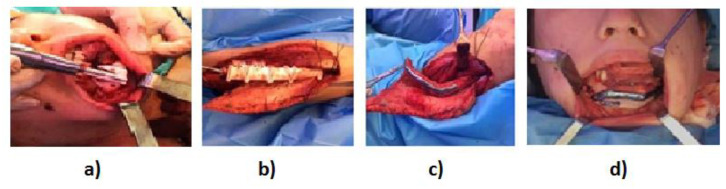
Details of the surgical procedure: (**a**) mandibular resection; (**b**) fibula flap modelling; (**c**) plate binding in the donor zone; and (**d**) positioning and binding of the flap in the mandibular defect.

**Table 1 cancers-15-02582-t001:** Clinical and surgical data of the 23 patients included in the study.

Variables	Number (%)
Men/women	13 (56.5)/10 (43.5)
Age, years, mean (SD)	52.8 (14.2)
Type of pathology	
Malignant	15 (65.2)
Benign	8 (34.8)
Histological type	
Oral squamous cell carcinoma	10 (43.5)
Odontogenic tumors (benign and malignant)	7 (30.4)
Sarcoma	2 (8.7)
Secondary deformity	1 (4.3)
Osteoradionecrosis	1 (4.3)
Infiltrating verrucous carcinoma	1 (4.3)
TNM stage of malignant tumors	15 (65.2)
T4N0	9 (39.1)
T4N1	3 (13.0)
T4N2a	3 (13.0)
Neoadjuvant treatment (RT or QT/RT)	4 (17.4)
Mandibular defect	
Type C (LCL, CL, LC, and CH)	13 (56.5)
Type H	4 (17.4)
Type L	6 (26.1)
Fibula skin flap positioning	
Intraoral internal	20 (87.0)
Extraoral external	3 (13.0)
Closure of the lower limb defect	
Direct	3 (13.0)
Skin graft	20 (87.0)
Postoperative complications (Clavien–Dindo)	
Grade I	7 (30.4)
Grade IIIb	4 (17.4)
Adjuvant treatment	12 (52.2)

SD: standard deviation; TNM: tumor node metastasis; T: tumor; N: node; RT: radiotherapy; CT: chemotherapy; type C: defect consisting of the entire central segment containing four incisors and two canines; LCL: lateral defect-to-bilateral angle defect; CL, LC: lateral angle-to-bilateral canines; CH: lateral segment defect including the condyle and central defect; type H: lateral defect of any length, including the condyle but not significantly crossing the midline; type L: defect of the same type without the condyle.

**Table 2 cancers-15-02582-t002:** Results obtained in the 12 single question domains of the UW-QOL questionnaire.

Domain	Patients Number	Mean (SD)	Median (Range)	% Best Score (of 100)	Importance of Domain *	Rank Order
Pain	21	86.9 (12.8)	75 (75–100)	48	10	6
Appearance	21	67.9 (19.6)	75 (25–100)	10	48	2
Activity	21	83.3 (16.5)	75 (50–100)	43	29	4
Recreation	21	84.5 (20.1)	100 (25–100)	52	5	7
Swallowing	21	84.5 (20.1)	100 (30–100)	67	10	6
Chewing	21	57.1 (39.6)	50 (0–100)	38	62	1
Speech	20	83.0 (19.5)	85 (30–100)	50	43	3
Shoulder	21	90.9 (18.4)	100 (30–100)	76	5	7
Taste	21	92.9 (13.1)	100 (70–100)	76	10	6
Saliva	21	78.1 (27.3)	100 (30–100)	52	24	5
Mood	20	82.6 (21.6)	87.5 (25–100)	50	24	5
Anxiety	20	87.5 (24.5)	100 (25–100)	70	10	6

* This asks about which three domain issues were the most important during the past 7 days, and results expressed as the percentage of patients choosing each domain.

**Table 3 cancers-15-02582-t003:** Responses to three global questions of the UW-QOL questionnaire.

Questions	Mean (SD)	% Best Scores *
A. Health-related QoL compared to month before had cancer	60.0 (34.8)	80
B. Health-related QoL during the past 7 days	73.3 (22.2)	81
C. Overall QoL during the past 7 days	71.4 (22.4)	81

Key to ratings: A: (0) much worse, (25) somewhat worse, (50) about the same, (75) somewhat better, (100) much better. B: (0) very poor, (20) poor, (40) fair, (60) good, (80) very good, (100) outstanding. C: (0) very poor, (20) poor, (40) fair, (60) good, (80) very good, (100) outstanding. * Best scores: A = % of scoring 50, 75, or 100; B and C = % scoring 60, 80, or 100.

**Table 4 cancers-15-02582-t004:** Mean scores of the 12 single question domains of the UW-QOL questionnaire.

Domain	First Author, Year [Reference] (Number of Patients)						
	Present Series (*n* = 21)	Li et al., 2014 [15] (*n* = 35)	Yang et al., 2014 [27] (*n* = 34)	Zhu et al., 2014 [25] (*n* = 33)	Luo et al., 2014 [28] (*n* = 32)	Zhang 2013 [14] (*n* = 31)	Wang 2009 [26] (*n* = 15)
Pain	86.9 (12.8)	82.2 (5.8)	67.4 (7.5)	76.4 (6.5)	80.6 (7.5)	87.6 (10.2)	86.7 (16.0)
Appearance	67.9 (19.6)	78.1 (11.6)	70.1 (6.6)	74.6 (9.6)	76.3 (8.7)	58.5 (2.1)	66.7 (29.4)
Activity	83.3 (16.5)	69.5 (7.6)	56.5 (9.1)	64.1 (8.3)	66.2 (9.1)	72.4 8.5)	76.7 (22.1)
Recreation	84.5 (20.1)	68.2 (10.6)	60.1 (9.1)	65.6 (8.7)	69.4 (7.1)	75.9 (6.1)	65.0 (33.8)
Swallowing	84.5 (20.1)	77.3 (6.8)	52.8 (9.0)	79.2 (7.2)	78.1 (5.1)	83.7 (1.6)	48.7 (26.9)
Chewing	57.1 (39.6)	28.5 (3.2)	33.1 (16.1)	32.4 (1.8)	30.3 (2.7)	42.2 (2.6)	36.7 (22.8)
Speech	83.0 (19.5)	71.3 (12.6)	55.3 (10.3)	68.8 (9.9)	66.4 (7.8)	47.9 (1.2)	53.3 (34.1)
Shoulder	90.9 (18.4)	80.3 (9.0)	65.9 (7.1)	81.1 (5.5)	82.3 (3.1)	92.4 (3.1)	82.0 (15.2)
Taste	92.9 (13.1)	71.2 (8.8)	55.6 (6.0)	80.5 (5.5)	78.7 (7.5)	90.3 (1.9)	80.7 (24.9)
Saliva	78.1 (27.3)	60.0 (7.6)	47.8 (8.9)	75.0 (9.7)	74.1 (8.0)	70.8 (1.5)	58.7 (28.2)
Mood	82.6 (21.6)	67.1 (1.2)	73.4 (11.5)	67.1 (1.2)	60.1 (3.0)	85.3 (7.9)	71.7 (31.1)
Anxiety	87.5 (24.5)	55.8 (8.2)	50.8 (14.3)	65.2 (8.6)	45.3 (9.6)	69.8 (6.3)	64.7 (66.7)

SD: standard deviation.

**Table 5 cancers-15-02582-t005:** Best scores obtained in the 12 single question domains of the UW-QOL questionnaire.

Domain	First Author, Year [Reference] (Number of Patients)				
	Present Series (*n* = 21)	Li et al., 2014 [15] (*n* = 35)	Zhu et al., 2014 [25] (*n* = 33)	Luo et al., 2014 [28] (*n* = 32)	Wang 2009 [26] (*n* = 15)
Pain	48	43	42	44	53
Appearance	10	26	36	31	20
Activity	43	9	6	NR	40
Recreation	52	0	3	6	33
Swallowing	67	29	49	28	7
Chewing	38	0	0	0	0
Speech	50	23	15	3	20
Shoulder	76	40	46	44	40
Taste	76	26	33	40	53
Saliva	52	42	42	22	13
Mood	50	11	9	3	40
Anxiety	70	0	0	6	40

Data as % best score (of 100) for each domain; NR: not reported.

**Table 6 cancers-15-02582-t006:** Importance of domain and rank order assigned to the 12 single question domains of the UW-QOL questionnaire.

Domain	First Author, Year [Reference] (Number of Patients)						
	Present Series (*n* = 21)	Li et al., 2014 [15] (*n* = 35)	Yang et al., 2014 [27] (*n* = 34)	Zhu et al., 2014 [25] (*n* = 33)	Luo et al., 2014 [28] (*n* = 32)	Zhang 2013 [14] (*n* = 31)	Wang 2009 [26] (*n* = 15)
Pain	10% (6)	0% (11)	5.9% (9)	0% (9)	0% (8)	7% (8)	7% (6)
Appearance	48% (2)	49% (3)	18% (7)	67% (2)	50% (3)	55% (3)	20% (5)
Activity	29% (4)	17% (7)	41% (4)	58% (3)	38% (4)	0% (11)	0% (8)
Recreation	5% (7)	14% (8)	0% (10)	15% (7)	13% (6)	0% (11)	0% (8)
Swallowing	10% (6)	6% (10)	47% (3)	0% (9)	3% (7)	13% (7)	93% (1)
Chewing	62% (1)	77% (1)	71% (1)	76% (1)	94% (1)	90% (1)	53% (2)
Speech	43% (3)	54% (2)	53% (2)	30% (4)	25% (5)	68% (2)	46% (3)
Shoulder	5% (7)	0% (11)	0% (10)	0% (9)	0% (8)	3% (9)	0% (8)
Taste	10% (6)	11% (9)	29% (5)	0% (9)	3% (7)	3% (9)	NR
Saliva	24% (5)	23% (5)	24% (6)	12% (8)	0% (8)	26% (4)	40% (4)
Mood	24% (5)	20% (6)	0% (10)	18% (6)	13% (6)	16% (6)	0% (8)
Anxiety	10% (6)	29% (4)	12% (8)	24% (5)	63% (2)	19% (5)	7% (6)

Data as percentage of patients choosing which three domains were the most important during the past 7 days. Rank order of domains in parenthesis; NR: not reported.

## Data Availability

Study data are available from the corresponding author upon request.

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
