# Peer review of "Quality of Life after Mandibular Reconstruction Using Free Fibula Flap and Customized Plates: A Case Series and Comparison with the Literature"

_cancers, 2023, doi:10.3390/cancers15092582_

Round 1

Reviewer 1 Report

This single-center retrospective study is very interesting, providing new data about the quality of life of Head and Neck Cancer patients. 

Although these data can be worth of publication, some methodological concerns lead me to require the authors some major revisions.

- Was a sample size calculation perfomed? 

- I found the statistical analysis inappropriate, since the study is probably underpowered to give any of the reported results

- The comparison with previous studies is very interesting, but I find inappropriate the statistical comparisons between such different studies.

Concluding, the results of the inferential statistical analysis should be removed, since it may be misleading for the readers. Further (multicentric?) studies should be performed in order to reach sound conclusion.

Reviewer 2 Report

Dear Authors,

It was a pleasure to conduct the review of the work entitled “Quality of Life After Mandibular Reconstruction Using Free Fibula Flap and Customized Plates: A Case Series and Comparison with Literature” in which the authors conducted a retrospective study about a very interesting and current topic for readers, providing suggestion for future research, but at current state it is not ready for publication in this notable journal. Some minor changes are needed to make it suitable for publication.

Below, you can find some considerations about the paper.

REVIEW REPORT

Broad comments

1.         Article Title

Is the title of the manuscript brief, appropriate, and indicative of the material which is contained in the manuscript? Yes.

2.         Abstract

Is the abstract concise? Yes.

Does it adequately describe the study? Yes.

Are the results and significances adequately presented? Yes.

3.         Introduction: Review of the Literature

Has the author cited the pertinent, but only the pertinent, literature? Yes, it appears so. 

Is the length of the introduction and the literature review appropriate or excessive? It is appropriate.

4.         Statement of Objectives

Is there a clear statement of the objectives of the study? Yes.

Are the objectives justified by the introduction? Yes.

5.         Description of Study Design: Material and Methods

Are the methods used in the study scientifically valid and technically correct? Yes.

Is the experimental group appropriate? There are some points that should be reviewed. About the inclusion criteria: Authors include both benign and malignant pathology. Nevertheless, these 2 groups of patients have different risk factors related to QoL (radiotherapy, chemotherapy, smoke, dental-related health status). The authors should conduct different analysis from these 2 groups of patients. 

Are the procedures described in sufficient detail for a clear understanding? Somehow. The procedures are described in a clear manner.

For studies of patient treatment, is the duration of the follow-up sufficiently long, and were the appropriate parameters of success or failure examined? Yes 

Were the outcome criteria identified and were they objectively and reliably measured? Somehow. To evaluate the impact to the appearance section, other methods are available (e.g., photogrammetry, cephalometric analysis, 3D-CT analysis). The authors could provide a paragraph in discussion section about these methods that could be used for esthetical analysis.

Other factors that could act as confounder of patient perception outcomes should be analyze: radiotherapy (impairment of salivary gland, trismus, mucositis, etc.,) oncological surgery (mouth opening limitation) oral healt status (dental extraction, edentulous patient) and other risk factors (smoke, age). These variables should be analyze in an adequate manner (also add some paragraph in discussion section)

6.         Statistical Analysis

Were the methods of statistical analysis appropriate for the study? Somehow. It should be implemented with statistical analysis of risk factors.

Did the author appropriately interpret significance and non-significance correctly? Yes.

Is review by a statistician needed?  Yes or No? No.

7.         Results

Are the results and data gathered in the study presented in a clear and logical method? Yes

Are tables and figures used to illustrate the data? Yes. 

8.         Discussion

Does the author explain the importance of his findings? Yes, in a clear way.

Are the results of this study discussed in light of other studies? Yes.

Are the comparisons with other studies appropriate and insightful? Yes.

9.         Conclusions

Are the conclusions consistent with the data and results presented in the manuscript? Yes.

Are the conclusions warranted by the results? Yes.

Are the conclusions overstated, too broad, or inappropriate, based on the data presented? No.

10.       Figures

Are the figures appropriate in number and clarity? Yes.

Should any of the figures be deleted or revised? No. Please provide high quality figures.

Are they appropriately cropped and labeled? Yes.

11.       References

Are the references current and accurate? Yes.

Are important references omitted? Yes. About the oral status of head and neck cancer patients.

If excessive, which should be deleted?  None.

12.       Grammar and Style

This article has not significant errors and its interpretation is easy to achieve. 

13.       General Comments

Some minor changes are needed to make it suitable for publication.

Specific comments

No specific comment referring to line numbers, tables or figures should be made.

Reviewer 3 Report

1.      The abstract should be broadened to give additional quantitative results.

2.      The present abstract was insufficient, please include the abstract's "take-home" message.

3.      Keywords is too much, choose the important please.

4.      The reviewer recommends to not using abbreviations in the keywords.

5.      What makes the author's novelty in the present article? My analysis suggests that other similar previous articles properly explain the points you have brought up in the current paper. Please be sure to emphasize anything truly novel in this work in the introductory section.

6.      To underline the submitted article gaps that the newest works tries to fill, it is crucial to explain the merits, novelty, and limits of earlier studies in the introduction.

7.      The authors needs to explain the advantages to titanium materials, it would be from corrosion resistance and biocompability aspect. In needs to explain and incorporated relevant reference as follows: Tresca Stress Simulation of Metal-on-Metal Total Hip Arthroplasty during Normal Walking Activity. Materials (Basel). 2021, 14, 7554. https://doi.org/10.3390/ma14247554

8.      Please note that last paragraph of the introduction section of this article should be explained the present article objectively.

9.      Additional figures in the introduction would improve the quality of the present article. Please provide it.

10.   To help the reader grasp the study's workflow more easily, the authors could include more visuals to the materials and methods section in the form of figures rather than sticking with the text that now predominates.

11.   What is the basis for patient selection? Is there any protocol, standard, or basis that has been followed? It is unclear since the patient is very heterogeneous with a small number. The resonance involved impacts the present result makes this study flaws. One major reason for rejecting this paper.

12.   It also is needed to include more information on tools, such as the manufacturer, the country, and the specification.

Round 2

Reviewer 1 Report

I would like to thank the authors for having taken into account my suggestion. 

The article is now suitable for publication.

Reviewer 3 Report

Reviewers greatly appreciate the efforts that have been made by the author to improve the quality of their articles after peer review. I reread the author's manuscript and further reviewed the changes made along with the responses from previous reviewers' comments. Unfortunately, the authors failed to make some of the substantial improvements they should have made making this article not of decent quality with biased, not cutting-edge updates on the research topic outlined. In addition, the author also failed to address the previous reviewer's comments, especially on comments number 5 (lack of novel), 6 (not captured state of the art), 7 (not incorporated literature), and 11 (the limitation of the research is critical flaws). Thank you very much for the opportunity to read the author's current work.